# Reclassification of ASFV into 7 Biotypes Using Unsupervised Machine Learning

**DOI:** 10.3390/v16010067

**Published:** 2023-12-30

**Authors:** Mark Dinhobl, Edward Spinard, Nicolas Tesler, Hillary Birtley, Anthony Signore, Aruna Ambagala, Charles Masembe, Manuel V. Borca, Douglas P. Gladue

**Affiliations:** 1United States Department of Agriculture, Agricultural Research Service, Foreign Animal Disease Research Unit, Plum Island Animal Disease Center, Orient, NY 11957, USA; mark.dinhobl@usda.gov (M.D.); edward.spinard@usda.gov (E.S.); nicolastesler2@gmail.com (N.T.); hillary.birtley@usda.gov (H.B.); 2United States Department of Agriculture, Agricultural Research Service, Foreign Animal Disease Research Unit, National Bio and Agro-Defense Facility, Manhattan, KS 66502, USA; 3Center of Excellence for African Swine Fever Genomics, Guilford, CT 06437, USA; anthony.signore@inspection.gc.ca (A.S.); aruna.ambagala@inspection.gc.ca (A.A.); charles.masembe@mak.ac.ug (C.M.); 4Oak Ridge Institute for Science and Education (ORISE), Oak Ridge, TN 37830, USA; 5National Centre for Foreign Animal Disease, Canadian Food Inspection Agency, Winnipeg, MB R3E 3M4, Canada; 6Department of Zoology, Entomology and Fisheries Sciences, School of Biosciences, College of Natural Sciences, Makerere University, Kampala P.O. Box 7062, Uganda

**Keywords:** African swine fever, ASFV, biotype, genotype, classification

## Abstract

In 2007, an outbreak of African swine fever (ASF), a deadly disease of domestic swine and wild boar caused by the African swine fever virus (ASFV), occurred in Georgia and has since spread globally. Historically, ASFV was classified into 25 different genotypes. However, a newly proposed system recategorized all ASFV isolates into 6 genotypes exclusively using the predicted protein sequences of p72. However, ASFV has a large genome that encodes between 150–200 genes, and classifications using a single gene are insufficient and misleading, as strains encoding an identical p72 often have significant mutations in other areas of the genome. We present here a new classification of ASFV based on comparisons performed considering the entire encoded proteome. A curated database consisting of the protein sequences predicted to be encoded by 220 reannotated ASFV genomes was analyzed for similarity between homologous protein sequences. Weights were applied to the protein identity matrices and averaged to generate a genome-genome identity matrix that was then analyzed by an unsupervised machine learning algorithm, DBSCAN, to separate the genomes into distinct clusters. We conclude that all available ASFV genomes can be classified into 7 distinct biotypes.

## 1. Introduction

The only member of the *Asfarviridae* family is the African swine fever virus (ASFV), which contains a large double-stranded DNA genome consisting of 150–200 genes. ASFV causes a severe disease in domestic swine and wild boar, African swine fever (ASF), resulting in economic losses in areas where ASF remains endemic or is causing outbreaks. Historically, ASFV has been characterized into 25 genotypes based on the partial sequencing of the *B646L* gene that encodes the major capsid protein p72. Although ASFV has been around for over 100 years [1], before 2007 the disease only sporadically left Africa. However, currently ASFV is causing a global pandemic, that started after ASFV was discovered in the Republic of Georgia in 2007 [2]. This outbreak has persistently spread across Europe, and Asia and, in 2021, reached the island of Hispaniola (Dominican Republic and Haiti) [3]. Just recently, in 2023, ASF made its first appearance in Sweden [4]. In the current pandemic, only genotype 2 has been detected outside of Africa, with the exception of China which has also detected a low virulent genotype 1 (that closely matches a historical vaccine strain) and a hybrid virus of these two strains [5]. Numerous reports have documented the existence of variations stemming from the genotype 2 strain whose origin can be traced back to Georgia. Genotype 2 variants have been identified in various regions across the globe, including Europe, Asia, Hispaniola, and Africa [3,6,7,8,9,10]. Indeed, some of these strains have mutations across the genome, genetic rearrangements, and deletions. To further complicate research efforts even more, recent reports indicate that, in Africa, ASFV isolates in domestic or smallholder farms have been restricted to only p72 genotypes 1, 2, 9, and 23 [2,7,11,12,13,14].

Subsequently, the significance and accuracy of the delimitations of genotypes have become a concern for the ASFV research community. Recently, during the Global African swine fever Research Alliance (GARA) meeting held in the Dominican Republic in May of 2022 and again at the GARA Gap analysis held in Uganda in February of 2023, the significance of p72 genotyping was discussed. ASFV genotyping based on the sequence of p72 was created with the purpose of the epidemiological tracking of the appearance of different field isolates, but its significance has been erroneously applied to other purposes, including prediction of cross-protection. Recently we analyzed all publicly available sequences for p72 and established criteria for genotyping, as this methodology is still in use in endemic and outbreak areas where the technologies for whole genome sequencing are likely to be unavailable [14]. We discovered that there were not 25 genotypes as previously reported, and after correcting some sequence analysis errors, we established a new criterion for p72 genotyping, demonstrating the existence of only 6 genotypes. 

With the recently approved vaccines in Vietnam for ASF against the current genotype 2 field strain, the question that arises is how many distinct ASFV genomes exist, and how many different vaccines will be needed to cover all current and future emerging strains of ASFV. With limited information about cross-protection of ASF vaccines, and since the only certain way to test cross-protection experimental evaluation in animal experiments, it is important as a starting point to group field isolates into distinct groups. This would facilitate the design of cross-protection studies and give a clear understanding of the landscape of circulating and historical ASFV strains. This methodology must also be implemented as evolutionary changes will create potential new field strains of ASFV which would need to be identified to determine as are true new emerging strains or if they fall within a cluster of previously identified ASFV strains. 

As reviewed by Qu et al. [15], efforts have been made to enhance the resolution of the p72 categorization of ASFV through the utilization of other genes, specifically p54 (E183L) and the central variable region (CVR) of B602L [15]. Still, due to the complexities inherent to the ASFV genome (large size, gain, and loss of genes, and hundreds of open reading frames—ORFs), the classification of ASFV based on small subsets of genes is inadequate.

Attempts to classify ASFV through whole genome analysis began in the 1980s utilizing restriction fragment length polymorphism [15,16,17]. This method facilitated the categorization of 23 isolates collected from Africa, Europe, and the Americas into five groups [15,17]. The recent advent of next generation sequencing (NGS) technologies has led to the assembly of whole ASFVs genomes, facilitating the phylogenetic analyses of ASFV by several research groups [18,19,20,21,22]. These analyses have resulted in the proposition that ASFV can be categorized into five distinct clades: alpha, beta, gamma, delta, and epsilon [20]. Still, categorization by whole genome analysis has its disadvantages: (1) Mutations in untranslated regions (UTR) are weighted the same as mutations that occur in ORFs, (2) synonymous mutations are weighted the same as nonsynonymous and nonsense mutations, and finally (3) mutations and deletions that occur in the highly variable genes within the MGF families are weighted the same as mutations that occur in conserved “core” proteins. Theoretically, one could partition and align each ORF, UTR, and the highly variable region (HVR) of the ASFV genome using a different evolutionary model for each region, however, this approach necessitates significant computational resources [15]. More recently, 41 ASFV genomes were analyzed using the chewBBACA pipeline [23,24]. In short, coding sequences (CDSs) over 250 nts long were extracted and binned by allele designation. Nucleotide sequences of alleles were then aligned, and a phylogenetic analysis was performed. While this methodology only examines mutations that occur in ORFs, the remaining disadvantages/challenges of whole genome nucleotide analysis remain. Finally, it should be noted that although phylogenetic analysis yields a visual representation of differences, the criteria employed to define clades may still be ambiguous.

In this study, we collected 220 non-duplicate whole genomes of ASFV isolated from NCBI. Genomes were annotated using ASFV Georgia 2007/1 as a reference and unidentified ORFs were annotated using CLC Genomics Workbench 23.0.2 (QIAGEN, Aarhus, Denmark). Annotations were manually curated, and translated to their predicted amino acid sequences, and homologous amino acid sequences encoded by the different genomes were aligned using MUSCLE [25] to create gene-level percent identity matrices via BioPython [26]. Gene-level percent identity matrices were weighted using an in-house developed algorithm and averaged to create a genome-genome percent identity matrix that was analyzed using the density-based clustering algorithm DBSCAN [27,28] to cluster the genomes based on similarity. Based on this research we propose that ASFV can be classified into seven biotypes.

## 2. Materials and Methods

The overall methodology used in this paper is documented in Figure 1 and further described in the sections that follow.

### 2.1. Description of the Dataset

The full-length genomes of 261 ASFV isolates were retrieved from the Nucleotide collection of NCBI on 1 June 2023 using the search term “African swine fever virus” and the following parameters: [porgn:_txid10497], Sequence length > 166,000 and Molecule types filter set to “genomic DNA/RNA”. Genomes from the following isolates SPEC_57, LIV_5_40, RSA_2_2008, Zaire, RSA_2_2004, and RSA_W1_1999 (GenBank accessions MN394630, MN318203, MN336500, MN630494, MN641877, and MN641876, respectively) were corrected of numerous SNPs using their raw sequencing data (Sequence Read Archive accessions SRR10282408, SRR10282646, SRR10282409, SRR10418876, SRR10418782, and SRR10418853, respectively).

### 2.2. Annotation of the Dataset

Using the isolate ASFV Georgia 2007/1 (GenBank accession: FR682468.2) as a reference, all genomes were annotated using the default parameters of Genome Annotation Transfer Utility (GATU) [29]. Overlapping annotations were manually corrected. In addition, the “Find Open Reading Frames” function of CLC Genomics Workbench 23.0 (Qiagen, Aarhus, Denmark) was used to detect ORFs that were not identified by GATU (minimum length = 110 codons). The identified ORFs were subsequently translated, compared to the NCBI database using the default parameters of BLASTP, and were assigned names based on their best match [30,31,32,33]. Any ORFs that matched a hypothetical protein that was not annotated in the Georgia 2007/1 genome were excluded from further analyses. In some instances, Georgia 2007/1 encodes a shortened version of a gene within one of the five multigene families (MGF) (MGF-100, MGF-110, MGF-300, MGF-360, and MGF-505). Accordingly, MGF annotations were manually extended to an earlier start codon if the start codon was over 100 nucleotides upstream and in-frame. The letters, starting with “a”, were added to gene names if a gene was split into multiple ORFs. Duplicate genes were indicated by the suffix “_1”. 

### 2.3. Curation of the Dataset

Annotations were translated to generate the predicted protein-coding sequences using CLC Genomic Workbench 23.0. To decrease computational time, 29 duplicate proteomes, (e.g., Pretoriuskop/96/4 (AY26136) and Pretoriuskop/96/4 (NC_044952)), were removed (Appendix A). Next, to correct for potential sequencing errors, multiple proteomes were removed: ASFV/Kyiv/2016/131 (MN194591) was excluded from further analysis as it contained numerous insertions and deletions (indels) resulting in frameshifts and early truncations in multiple well-conserved proteins, Arm/07/CBM/c4 (LR881473) was excluded since it originated from a mixed stocked [34] and finally 10 proteomes with more than 50 ambiguous nucleotides (MH910495, MH910496, MW788405, MW788407, MW788408, MW788409, MW788410, MW788411, OK236383, ON402789) were removed. To further refine the dataset, corrections were applied to individual proteins; 545 proteins with ambiguous amino acids or less than 5 homologues of the same length (excluding genes of the MGF families) were removed. After curation, the final dataset consisted of 220 genomes encoding a total of 242 unique genes (Appendix A). 

### 2.4. Protein Alignment

The 242 homologues were aligned using MUSCLE v 3.8.31 [35] using a gap extension penalty of −1.0, and a gap opening penalty of −10.0. The protein alignments were then converted to gene-level percent identity distance matrices using Biopython [26].

### 2.5. Genome Level Analysis and Clustering

Weights were designed to give more weight to “core” genes (non-MGF, non-ACD, and non-hypothetical genes) (Appendix A) as well as genes present in more genomes. Genes were assigned a weight using the following equation:$$\begin{array}{c} W_{Gene}=C\times \sqrt{\#\;of\;genomes\;encoding\;gene}\\ C=1\;\mathrm{if}\;\mathrm{core},\frac{1}{4}\;\mathrm{if}\;\mathrm{not}\;\mathrm{core} \end{array}$$

It should be noted that genomes that did not encode a gene would not have their average impacted by the absence of said gene. Using the weighted average, gene-level percent identity distance matrices were then averaged into a single cumulative genome-genome percent identity matrix. This final percent identity matrix was run through the spatial clustering machine learning algorithm DBSCAN from scikit-learn 1.3.0 [27,36], 100 times using an Epsilon (eps) value of 0.01 to 1 at intervals of 0.01 and the following constant parameters: min_samples = 1, metric = ‘euclidean’, metric_params = None, algorithm = ‘auto’, leaf_size = 30, and *p* = None. The process is summarized in Figure 2.

## 3. Results

### 3.1. Full-Length Genomes on NCBI

All 261 full-length ASFV genome sequences were downloaded from NCBI and processed as described in the Materials and Methods Section 2 (Appendix A) resulting in a database of 220 curated genomes. ASFV annotation and gene nomenclature have not been standardized and have resulted in some genes having multiple alternative names [37], reviewed on https://asfvgenomics.com/proteindatabase (accessed on 1 December 2023). To avoid potential conflicts in nomenclature, all genomes were re-annotated using Georgia 2007/1 (GenBank accession: FR682468.2) as a reference, resulting in 220 non-redundant genomes encoding an average of 175.9 genes (±15.1). 

### 3.2. Clustering of ASFV

Pairwise percent identities were calculated for each gene. Evaluating virus proteins encoded in the central region of the genome, the following proteins had the lowest percent identities (considering at least 100 genomes): A118R (0.73 ± 0.292), EP153R (0.77 ± 0.22), EP402R (0.79 ± 0.212), A238L (0.83 ± 0.155), L60L (0.84 ± 0.171), DP71L (0.89 ± 0.264), A240L (0.91 ± 0.1), I10L (0.92 ± 0.08), I196L (0.92 ± 0.093), L11L (0.93 ± 0.094) (Appendix A). Following their calculation, all pairwise percent identity matrices were averaged into a single identity matrix using an algorithm developed in-house as described in the Materials and Methods section. Our weighting matrix increased the value of (1) genes that were consistently encoded and (2) 123 conserved “core” proteins (Appendix A). Genes annotated as “ACD ####” were given less weight because they are hypothetical proteins with no known function. Genes within the five MGF families (MGF100, MGF110, MGF300, MGF360, and MGF505) were given less weight since they are highly variable and often contain homopolymer stretches of G/C or A/T that can result in indels that lead to deletions, truncations, and fusions [10,22] and reviewed in [37]. The weighted gene-level percent identity distance matrices were then averaged into a single genome-genome percent identity matrix, and clustered based on similarity using DBSCAN. Clustering refers to the process of dividing a dataset into subsets of points. The aim is to group similar points together in the same cluster while separating dissimilar points into different clusters. The utilization of DBSCAN as a clustering technique offers an extra level of dependability, as DBSCAN identifies clusters of arbitrary shape and does not require a specific number of clusters to be specified [36]. Accordingly, it is more powerful than other clustering methods such as k-means which place data in clusters based on their proximity to a central point and require a specific number of clusters to be identified before analysis. The DBSCAN algorithm produces an output whereby each genome is assigned a numerical value ranging from 0 to *n*, where *n +* 1 = the total number of clusters at a given epsilon (Ɛ) value. At each Ɛ level, genomes that clustered based on similarity were assigned an identical number. DBSCAN was executed for 100 iterations of the parameter Ɛ, ranging from 0.01 to 1, with an increment of 0.01 for each iteration (Appendix A). Ɛ denotes the radius of a neighborhood centered at a point x. In simpler terms, as the value of Ɛ increases, the maximum number of dissimilarities permitted within a cluster also increases, leading to a reduction in the overall number of unique clusters.

Figure 3 shows the results of DBSCAN, highlighting the number of clusters and how they merge at Ɛ = 0.05, 0.06, 0.07, 0.09, 0.10, 0.12, 0.13, 0.14, 0.21, and 0.32 until total convergence occurs at epsilon = 0.53. At the lowest epsilon = 0.05, there were 18 groups, 10 of which (TAN/08/Mazimbu (ON409981), Malawi Lil-20/1 (AY261361), Ken05/Tk1 (KM111294), Uvira B53 (MT956648), BUR/18/Rutana (MW856067), Uganda (Unpublished), RSA/2/2004 (MN641877), Tengani 62 (AY261364), Mkuzi 1979 (AY261362), LIV 5/40 (MN318203) were composed of only a single genome. The remaining eight groups were named after a prototypical genome (Ken06, Kenya 1950, Warmbaths, Warthog, Benin, K, Recombinant, and Georgia) and were composed of 8, 2, 3, 4, 66, 2, 3, and 122 genomes, respectively (Appendix A). 

The approximate percent similarity and amino acid changes that constitute each cluster at each of the given Ɛ values were estimated by comparing Benin 97/1 (AM712239) to a newly clustered isolate (Table 1). As expected, as Ɛ increases, the weighted percent similarity within each group decreases and the total number of amino acid differences increases. Compared to the previous standard of ASFV classification, genotyping by p72 (*B646L*), the weighted similarity metric exhibits significantly lower values compared to p72 similarity at each Ɛ value. This suggests that the accumulated differences that contribute to the weighted similarity metric are more responsive and discerning in terms of classification than relying solely on p72 similarity alone. 

### 3.3. ASFV Can Be Classified as 7 Biotypes

At three epsilon values between 0.05 and 0.10 (0.06, 0.07, and 0.09) the 18 clusters merge based on similarity until forming 7 clusters at epsilon = 0.10 (Figure 3). Larger epsilon values (0.12, 0.13, 0.14, 0.21, and 0.32) continued to group the isolates, decreasing the total number of clusters (6, 5, 4, 3, and 2, respectively) until all genomes converged into a single cluster at an epsilon value of 0.53. Accordingly, for our new classification, we chose a cutoff value of epsilon = 0.10 as it was large enough to combine all historic genotype I isolates, yet small enough that genotype I isolates remained separated from the recently described recombinants which are composed of genetic sequences derived from NHV (genotype 1) and ASFV-G (genotype 2) [5]. We propose that this new grouping be referred to as biotypes. 

The 7 biotypes present when epsilon = 0.10 have some similarities to the traditional classification of ASFV based on p72 sequencing (Table 2 and Appendix A) [38]. Biotype 1 is composed of 69 isolates that would traditionally be considered part of Genotype I (Group Benin, Group K, and isolate LIV 5/40,). However, Mkuzi 1979 (AY261362), whose genotype has been ambiguous, historically classified as genotype I, VII, or XII, was also included in this biotype [14,18,39]). Interestingly, isolates within biotype 1 have been collected from outbreaks that occurred in many different regions including eastern Africa (Group K), southern Africa (LIV 5/40 and Mkuzi 1979), western Africa (Benin 97/1 and Ghana2021-95), Europe (NHV, L60, E75, OURT 88/3, E75, and the Sardinia strains), Asia (Pig/HeN/ZZ-P1/2021 and Pig/SD/DY-I/2021) and the island of Hispaniola (DR-1980) and that have spanned multiple decades (1949 to 2021) (Table 3). 

Biotype 2 is composed of 122 genotype II isolates that are derivatives of the current pandemic strain ASFV Georgia 2007/1. It also included the recent isolates YNFN202103 (ON400500), Nigeria-RV502 (OP672342), and Ghana2022-35 (OP479889), which all contain a nearly 6.5 kilobase pair deletion which resulted in the loss of 14 ORFs (MGF 110-8L, MGF 100-1R, ACD 00190, MGF 110-9L, ACD 00210, MGF 110-10L-14 L, ACD 00240, MGF 110-12L, MGF 110-13La, MGF 110-13Lb, ACD 00270, MGF 360-4L, ACD 00300 and ACD 00350) [10]. Since all the lost ORFs are annotated as an MGF family or hypothetical proteins, which are given less weight by the algorithm, and because the algorithm does not punish genomes for not encoding genes, YNFN202103, Nigeria-RV502, and Ghana2022-35 grouped within biotype 2.

Biotype 1–2 Recombinant consists of 3 recently isolated genomes from China. The genomes consist of biotype 1 and biotype 2 sequences and were believed to be the result of a recombination between NHV (also known as ASFV/NH/P68) (KM262845) and Georgia 2007/1 [5,39]. 

Biotype 3 consists of 9 unique genomes that were all isolated from southern Africa between 1962 and 2008. Recent reanalysis of the p72 genotypes grouped these isolates into genotype 2 with the derivatives of Georgia 2007/1 [14]. However, full genome analysis clearly indicates that the genomes are different when compared to Georgia 2007/1. Further, large epsilon values (Ɛ = 0.14) cluster biotype 3 with biotype 1 before merging with biotype 2 (Ɛ = 0.21). Interestingly, four isolates (Warmbaths (AY261365), RSA/2/2008 (MN336500), SPEC 57 (MN394630), and Pretorisuskop/96/4 (AY261363)) were collected from a tick, three isolates were collected from a warthog or wild boar (RSA_2_2004 (MN641877), Warthog (AY261366), and RSA/W1/1999 (MN641876)), while only two isolates (Zaire (MN630494) and Tengani 62 (AY261364)) were collected from domesticated pigs. Taken together, the metadata suggests the genetic adaption of ASFV to ticks or different host species [40,41]. Still, care must be taken not to overanalyze the results as other strains isolated from ticks in South Africa (LIV 5/40 (MN318203), Mkuzi (AY261362), and Malawi Lil-20/1 (ON409981)) did not group into biotype 3.

The three remaining biotypes are exclusively made up of isolates collected from East Africa. Biotype 4 is composed of 6 unique genomes (Ken05/Tk1 (KM111294), Kenya 1950 (AY261360), ASFV Ken.rie1 (LR899131), Uvira B53 (MT956648), BUR/18/Rutana (MW856067), Uganda (unpublished)) which were historically categorized as genotype X. Interestingly, although Ken05/Tk1 was historically characterized as genotype X based on its nucleotide sequences, its predicted p72 protein sequence is identical to the protein sequence of a genotype IX. Biotype 5 is composed of Ken06.Bus (KM111295), R8 (MH025916), R7 (MH025917), R25 (MH025918), N10 (MH025919), R35 (MH025920), TAN/16/Magu (ON409980), Ken1033 (unpublished) which were historically categorized as genotype IX. Biotype 6 consists of Malawi Lil-20/1 (1983) (AY261361) and TAN/08/Mazimbu (ON409981) which were historically classified as genotype VIII and XV.

### 3.4. Webportal for Automatic ASFV Biotyping and Genotyping

A tool has been provided on the Center of Excellence for Swine Fever Genomics website (https://asfvgenomics.com/upload) (Accessed on 1 December 2023) that analyzes a novel ASFV genome uploaded by the user and returns the most likely biotype (manuscript in submission). Moreover, as genotyping is still widely recognized and as a method to connect contemporary variants with past samples the closest p72 match will be predicted Additionally, the tool will issue warnings for highly unlikely genomes (indicating a potential need for reassembly). In the future, a function will be implemented that will detect new potential biotypes or groups—if a new potential biotype or group is detected, an email address will be given so that we can correspond about results and aid in the classification of your genome. 

## 4. Discussion

Historically, p72 (*B646L*) genotyping was conducted for the purpose of disease tracking and, because of the lack of next-generation sequencing technologies in many regions impacted by ASF [14], continues to be used to classify ASFV isolates. While a novel p72 classification scheme has reduced the number of genotypes from 25 to 6 and set clear parameters to define groupings within a genotype [14], it is clear that classification based on a single gene is insufficient for categorizing a virus as complex as ASFV. Any classification that is exclusively based on p72 is ultimately dependent on a handful of amino acid changes and accordingly, is easily prone to error as even a single sequencing error can result in a severe misclassification or the generation of a new genotype. As indicated in Table 1, using a method that considers more than just p72, the classification of ASFV can instead be based on hundreds or thousands of amino acid changes, avoiding the problems associated with genotyping by p72 alone. For example, while Warmbaths, Tengani 62, Pretorsuskop/96/4 and ASFV Georgia 2007/1 encode an identical p72 protein sequence, examination of the rest of the proteome reveals that ASFV Georgia 2007/1 is the outlier and far less similar. Conversely, while Ken05/Tk1 and Kenya 1950 encode a different p72, examination of the rest of the proteome reveals they are more similar. 

Other groups have attempted to classify 60 full-length ASFV genomes into clades based on the longest common sequence (LCS) methodology [20]. While this method is better than classifying ASFV solely based on the partial sequencing of p72, we believe our strategy is superior as it instead relies on a gene-level comparison that ignores non-coding regions, examines amino acid changes rather than nucleotide changes, weighs the results based on the number of times a gene is represented and whether the gene is a “core” protein, and was able to analyze the genomes of 220 ASFV isolates. 

A comparison of results based on historical genotyping, our corrected p72 genotyping [14], clades [20], and our biotyping are shown in Table 2. Note, that as a result of the incomplete sequencing of their genomes, it was not possible to conduct biotyping analysis on the genotype 23 isolates. All the latter three methods reduce the number of groups from over twenty to less than eight. Further, both biotypes and clades suggest that typing by p72 alone is insufficient—both historical and corrected p72 genotypes contain multiple biotypes/clades. For example, genotype 2 includes biotypes 1–2R, 2, and 3; it also includes clades Beta, Epsilon, and Delta. Lastly, biotypes and clades were also similar in that genotype 1 isolates collected from outbreaks in Europe and Africa grouped together (biotype 1 and alpha clade) and all genotype 2 isolates grouped within the same biotype and clade (biotype 2 and beta clade). 

Differences between groups created from the biotype and clades methodology were observed for certain isolates: (1) Tengani 62; our biotyping method groups it with the other strains originating from southern Africa (Warmbaths and Warthog), while it is the sole member of the delta clade. (2) The Eastern African groups (biotypes 4–6) were grouped into a single clade (gamma). (3) Liv 5/40 is classified along with Warmbaths and Warthog in the clades manuscript; however, for this analysis Liv 5/40 was reassembled and grouped with Mkuzi1979 within the biotype 1 group. Of course, at the time of publication certain genomes, such as those within biotype 1–2 Recombinant and TAN/08/Mazimbu, had yet to be sequenced and/or isolated and could not be analyzed. 

Care should be taken when interpreting the biotypes since it was not the intention of this manuscript to separate isolates based on virulence. Previous studies have analyzed ASFV virulence by examining the presence or absence of functional domains encoded by attenuated and virulence strains [42]. In its current iteration, both an attenuated and the virulent parental strain could be clustered together in the same biotype as was observed for BA71 and BA71V, K49 and KK262, L60 and OURT 88/3, and L60 and NHV. 

In the future, as the assessment of potential coverage induced by various vaccines would necessitate cross-protection studies, we propose utilizing biotypes as a means to determine what strains are more closely related, since biotyping is based on the analysis of the entire genome, rather than on a single ASFV protein. As these studies are performed there will be a better understanding of the serotype or biotype that specifically correlates with protection against ASFV. Accordingly, the utilization of a clustering algorithm-based methodology, as opposed to a phylogenetic tree, for the classification of ASFV, enables the biotyping classification to be modified in response to future data. In addition, as ASF continues to have prolonged outbreaks and with the increasing number of ASFV isolates being fully sequenced by next-generation sequencing, it is possible that novel and highly heterogeneous variants of ASF could be found, where it may be necessary to adjust the epsilon value to classify ASFV into a greater or lesser number of biotypes. Further, as many ASFV strains, such as the isolates that make up genotype 23, ETH/AA (KT795353), ETH/017 (KT795355), ETH/1 (KT795354), ETH/004 (KT795356), ETH/2a (KT795358), and ETH/1a (KT795359), have only been partially sequenced and could not be analyzed, the number of biotypes may expand as more historic isolates are fully sequenced. Moreover, with our current knowledge along with the development of a web-based tool to easily identify the biotype of ASFV, we believe standardization of ASFV isolates by biotypes is possible and constitutes the most accurate classification for ASFV. 

## Figures and Tables

**Figure 1 viruses-16-00067-f001:**
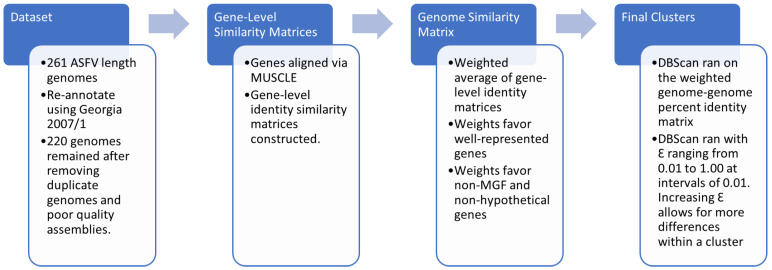
Summary of the analysis pipeline described in Materials and Methods.

**Figure 2 viruses-16-00067-f002:**
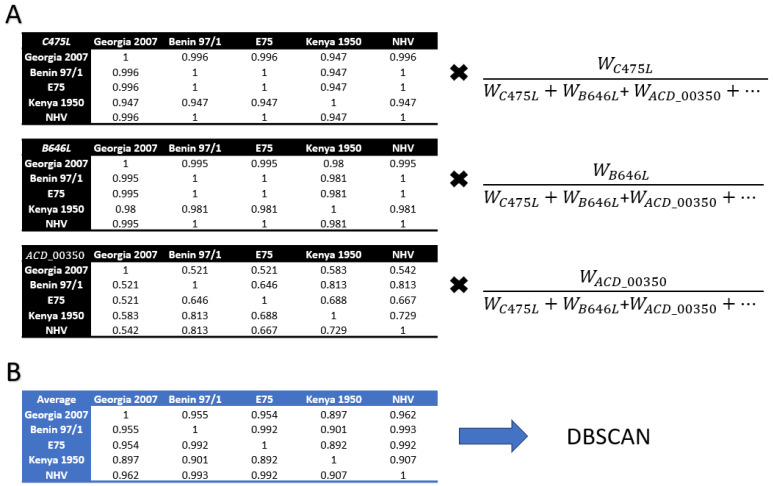
Process for going from pairwise percent identity gene-gene identity matrices to DBSCAN. (**A**) Gene-level identity matrices are multiplied by their weights and averaged resulting in the (**B**) genome-genome identity matrix. The average genome-genome identity matrix is then analyzed via DBSCAN to identify clusters based on similarity. The weight equation is described in Section 2.4.

**Figure 3 viruses-16-00067-f003:**
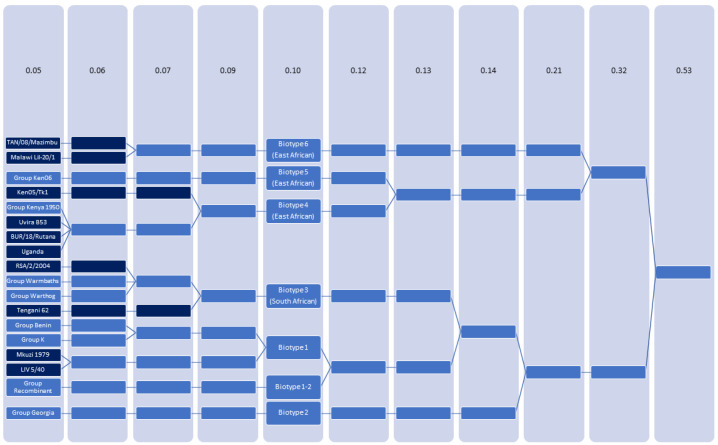
Results of DBSCAN demonstrate the number of clusters at the indicated epsilon value. Brackets indicate the convergence of clusters. Dark blue boxes indicate a sequence from a single isolate, and medium blue boxes indicate a group of isolates with similar sequences.

**Table 1 viruses-16-00067-t001:** Example Differences Between Benin 97/1 and the Indicated Isolate at the Given Epsilon (Ɛ) Level.

Ɛ	Example Isolate	Weighted Similarity	p72 Similarity
0.05	NHV (KM262845)	0.99313 (412)	1.00000 (0)
0.07	K49 (MZ202520)	0.98256 (1046)	1.00000 (0)
0.14	Warmbaths (AY261365)	0.96428 (2143)	0.99536 (3)
0.21	ASFV Georgia 2007/1 (FR682468)	0.95485 (2709)	0.99536 (3)
0.53	Malawi Lil-20/1 (AY261361)	0.92482 (4511)	0.98297 (11)

Parenthetical numbers indicate the estimated number of amino acid changes compared to the reference.

**Table 2 viruses-16-00067-t002:** Comparison of biotypes, p72 genotypes, historic genotype, and clades by group.

Group	Biotype	Updated Genotype ^1^	Historic Genotype ^2^	Clade ^2^
Group Benin	1	1	I	Alpha
Mkuzi 1979 (AY261362)	1	1	I/VII ^3^	Alpha
LIV 5/40 (MN318203)	1	1	I/VII	Epsilon ^4^
Group K	1	1	I	N/A
Group Recombinant	1–2 R	2	II	N/A
Group Georgia	2	2	II	Beta
RSA/2/2004 (MN641877)	3	2	XX	N/A
Group Warmbaths	3	2	III	Epsilon
Group Warthog	3	2	IV	Epsilon
Tengani 62 (NC_044951)	3	2	V	Delta
Ken05/Tk1 (KM111294)	4	9	X ^5^	Gamma
Uvira B53 (MT956648)	4	9	X	N/A
BUR/18/Rutana (MW856067)	4	9	X	N/A
Uganda (UnPub002)	4	9	X	N/A
Group Kenya 1950	4	9	X	Gamma
Group Kenya 06	5	9	IX	Gamma
Malawi Lil-20/1 (1983) (NC_044954)	6	8	VIII	Gamma
TAN/08/Mazimbu (ON409981)	6	15	XV	N/A

^1^ Referenced from [14]. ^2^ Referenced from [20]. ^3^ Categorized as I or VII [20]. ^4^ For this manuscript, Liv5/40 was reassembled, possibly explaining a difference in groups. ^5^ Ken05/Tk1 was classified as genotype X, however its p72 protein sequence is identical to IX genotypes. N/A represents there was no Clade information available.

**Table 3 viruses-16-00067-t003:** Biotype classification of ASFV isolates.

Biotype	Isolate
1	15998, 22649, 23221, 30322, 31208, 34403, 44076, 46830, 47039, 51268, 53706, 56140, 74377, 98039, 103917/18, 139/Nu/1981, 140/Or/1985, 141/Nu/1990, 142/Nu/1995, 19155_WB, 2019 WB, 22653/Ca/2014, 22943_2008, 26/Ss/2004, 26544/OG10, 28784WB, 33262WB, 33747 WB, 47/Ss/2008, 49179 WB, 4996 WB, 55234/18, 56/Ca/1978, 57/Ca/1979, 60/Nu/1997, 63525 WB, 7212WB, 72398 WB, 72407/Ss/2005, 72912 WB, 85/Ca/1985, 97/Ot/2012, BA71, BA71V, Benin 97/1, Ca1978_2, DR-1980, E75, Ghana2021-95, K49, KK262, L60, LIV/5/40, LO2018 major, LO2018 minor, Mkuzi 1979, NHV, Nu1979, Nu1986, Nu1990_1, Nu1991_2, Nu1991_3, Nu1991_7, Nu1993_2, Nu1995_3, Or_1984, Or1993_1, OURT 88/3, Pig/HeN/ZZ-P1/2021, Pig/SD/DY-I/2021
1–2 Recombinant	Pig/Henan/123014/2022, Pig/Inner Mongolia/DQDM/2022, Pig/Jiangsu/LG/2021
2	2020ASP01832, 2020ASP02103, 2020ASP02805, 2020ASP02894, 2021ASP00484, 2021ASP00703, 2021ASP00902, 2021ASP00921, 2021ASP01917, 2021ASP01919, 2021ASP01957, 2021ASP02148, 2021ASP02207, 2021ASP02665, 2021ASP03144, 2021ASP03251, 2021ASP03380, 2021ASP03384, 2021ASP03643, 2021ASP03658, 2021ASP03711, 2021ASP03740, 20355/RM/2022_Italy, 2802/AL/2022 Italy, A4, ABTCVSCK_ASF001, ABTCVSCK_ASF007, AQS-C-1-21, AQS-C-1-22, AQS-P-201202, AQS-P-20901-1, Arm/07/CBM/c2, ASF-MNG19, ASFV Belgium 2018/1, ASFV CzechRepublic 2017/1, ASFV Georgia 2007/1, ASFV Germany 2020/1, ASFV Korea/pig/Yeoncheon1/2019, ASFV Wuhan 2019-1, ASFV/Amur 19/WB-6905, ASFV/ARRIAH/CV-1/30, ASFV/ARRIAH/CV-1/50, ASFV/Kabardino-Balkaria 19/WB-964, ASFV/Kaliningrad_17/WB-13869, ASFV/Kaliningrad_18/WB-12516, ASFV/Kaliningrad_18/WB-12523, ASFV/Kaliningrad_18/WB-12524, ASFV/Kaliningrad_18/WB-9734, ASFV/Kaliningrad_18/WB-9735, ASFV/Kaliningrad_19/WB-10168, ASFV/Korea/pig/PaJu1/2019, ASFV/LT14/1490, ASFV/pig/China/CAS19-01/2019, ASFV/POL/2015/Podlaskie, ASFV/Primorsky 19/WB-6723, ASFV/Primorsky_19/DP-8235, ASFV/Timor-Leste/2019/1, ASFV/Ulyanovsk 19/WB-5699, ASFV/Zabaykali/WB-5314/2020, ASFV/Zabaykali_20/DP-4905, ASFV_Hanoi_2019, ASFV_NgheAn_2019, ASFV-SY18, ASFV-wbBS01, ASFV-wbShX01, Belgium/Etalle/wb/2018, CADC_HN09, China/GD/2019, China/LN/2018/1, CN/2019/InnerMongolia-AES01, Estonia 2014, Ghana2022-35, GZ201801, GZ201801_2, HB03A, HB31A, HuB20, IND/AR/SD-61/2020, IND/AS/SD-02/2020, JX21, Kashino 04/13, Korea/HC224/2020, Korea/YC1/2019, LYG18, MAL/19/Karonga, Nigeria-RV502, Odintsovo_02/14, OP823268, OP823269, Pig/Heilongjiang/HRB1/2020, Pig/HLJ/2018, Pol16_20186_o7, Pol16_20538_o9, Pol16_20540_o10, Pol16_29413_o23, Pol17_03029_C201, Pol17_04461_C210, Pol17_05838_C220, Pol17_31177_O81, Pol17_55892_C754, Pol18_28298_O111, Pol19_53050_C1959/19, SY-1, SY-2, TAN/17/Kibaha, TAN/17/Mbagala, TAN/20/Morogoro, Tanzania/Rukwa/2017/1, Vietnam/Pig/RG-1/01, Vietnam/Pig/RG-2/01, Vietnam/Pig/RG-3/01, Vietnam/Pig/RG-4/01, Vietnam/Pig/RG-5/01, Vietnam/Pig/RG-6/01, Vietnam/Pig/RG-7/01, VN/HY-ASFV1(2019), VN/QP-ASFV1(2019), VNUA-ASFV-05L1/HaNam/VN/2020, VNUA-LAVL2, wild boar/SNJ/2020, Yangzhou, YNFN202103,
3	Pretoriuskop/96/4, RSA/2/2004, RSA/2/2008, RSA/W1/1999, SPEC_57, Tengani 62, Warmbaths, Warthog, Zaire
4	ASFV Ken.rie1, BUR/18/Rutana, Ken05/Tk1, Kenya 1950, Uganda, Uvira B53,
5	Ken06.Bus, Ken1033, N10, R25, R35, R7, R8, TAN/16/Magu,
6	Malawi Lil-20/1, TAN/08/Mazimbu

## Data Availability

All data is publicly available as referenced in the manuscript.

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
