# Peer review of "Reclassification of ASFV into 7 Biotypes Using Unsupervised Machine Learning"

_viruses, 2023, doi:10.3390/v16010067_

Round 1

Reviewer 1 Report

Comments and Suggestions for Authors

Dinhobl et al. described the development of a novel platform (https://asfvgenomics.com/upload) for classifying the African swine fever virus (ASFV) based on all the encoded proteins rather than a single p72 protein. 220 reannotated ASFV genomes were analyzed using an unsupervised machine learning algorithm, DBSCAN. The authors demonstrated that all current ASFV genomes could be classified into 7 distinct Biotypes. Moreover, the authors compared these 7 Biotypes with p72 genotypes and historic genotypes, which showed apparent differences. Overall, the manuscript is interesting and informative, and the study is well-designed. The new classification of ASFV based on the whole genome is logical and practical. 

1. The major concern of this reviewer is that the ASFV genome is around 190 kb in length; in contrast, the p72 protein has only 646 amino acids. The sizes between them are huge; therefore, the difference between these two classification methods is no wonder (Table 2). Thus far, most of the available ASFVs were sequenced to cover p72 but not entire genomes, which would limit the application of this novel ASFV classification method targeting the full-length genome. Did the authors consider developing a platform using a similar algorithm focusing on the p72 and comparing it with the current traditional methods?

2. The authors concluded that the known-to-date ASFV genomes can be classified into 7 distinct Biotypes. Is there any biological significance between these 7 Biotypes? Or is the classification just based on the sequence similarities between ASFV genomes? Curiously, why the authors name them Biotypes?

3. As there are only several ASFV genomes in Biotypes 3 to 6, are these viruses in each Biotype epidemiologically associated with each other?

4. Presumably, the ASFV strains with partial genomic sequences may also present new Biotypes when their full-length genome is sequenced, indicating that the classified ASFV Biotypes will continue to expand. The authors should also discuss this aspect.

5. Results, 3.1: Please specify the number of the full-length ASFV genome sequences

6. Figure 3: The letters are nonreadable. Suggest increasing font size.

7. Some References are not well-organized. For example, a ref is lacking in the Introduction, para 1, line 12; ref style is inappropriate in Introduction, para 5, line 7 and para 6, line 7; note in the second line below Table 2; a blank line is in ref 13 in References. 

Author Response

  1. The major concern of this reviewer is that the ASFV genome is around 190 kb in length; in contrast, the p72 protein has only 646 amino acids. The sizes between them are huge; therefore, the difference between these two classification methods is no wonder (Table 2). Thus far, most of the available ASFVs were sequenced to cover p72 but not entire genomes, which would limit the application of this novel ASFV classification method targeting the full-length genome. Did the authors consider developing a platform using a similar algorithm focusing on the p72 and comparing it with the current traditional methods?

Answer: We completely agree with the reviewer that classifcation based on p72 vs an entire genome will yield completely different results. We recently performed a similar analysis with p72 (Spinard, E.; et al A Re-Evaluation of African Swine Fever Genotypes Based on p72 Sequences Reveals the Existence of Only Six Distinct p72 Groups), however we did not feel it was appropriate to utilize machine learning to group the different p72 isolates when a simple cut off of a maximum difference of 2 amino acids was enough to seperate the groups.

  1. The authors concluded that the known-to-date ASFV genomes can be classified into 7 distinct Biotypes. Is there any biological significance between these 7 Biotypes? Or is the classification just based on the sequence similarities between ASFV genomes? Curiously, why the authors name them Biotypes?

Answer: All ASFV experts will agree that there is a large gap of information for ASFV particularly in which proteins have functional significance when mutated for the pathogenesis ASFV. This information is likely 20+ years away and a correct classification of ASFV is desperately needed. Classification in this paper is based off of similairty between proteins that are predicted to be encoded in each genome. The term Genotype is already widley used within the ASFV to refer to classification based on the p72 gene and the term clades has also been used. We felt that biotype was appropriate since it can mean ”a strain distinguished from other microorganisms of the same species by its physiological properties.”

  1. As there are only several ASFV genomes in Biotypes 3 to 6, are these viruses in each Biotype epidemiologically associated with each other?

Answer: Groups 4-6 were predominately collected from Western Africa and Biotype 3 was predominately collected from Southern Africa and have not been found outside of these regions. Still, Biotypes 1 and 2 have been found in both regions. The disparity in the number of genomes is more so due to the occurrence of two pandemics that have occurred, namely the one in the 1960s and the ongoing pandemic that started in 2007. Biotypes 1 and 2 exhibit a notably higher count of genomes as the progenitor strains of these Biotypes have triggered outbreaks beyond the African continent with the most recent outbreak reaching Europe, Asia, and the island of Hispaniola.

  1. Presumably, the ASFV strains with partial genomic sequences may also present new Biotypes when their full-length genome is sequenced, indicating that the classified ASFV Biotypes will continue to expand. The authors should also discuss this aspect.

Answer: We have revised the following paragraph to better illustrate this idea:

“In addition, as ASF continues to have prolonged outbreaks and with the increasing number of ASFV isolates being fully sequenced by next-generation sequencing, it is possible that novel and highly heterogeneous variants of ASF could be found, where it may be necessary to adjust the epsilon value to classify ASFV into a greater or lesser number of Biotypes. For example, the isolates that make up genotype 23, ETH/AA (KT795353), ETH/017 (KT795355), ETH/1 (KT795354), ETH/004 (KT795356), ETH/2a (KT795358), and ETH/1a (KT795359), could not be analyzed since their genomes have not been fully sequenced.”

To

“In addition, as ASF continues to have prolonged outbreaks and with the increasing number of ASFV isolates being fully sequenced by next-generation sequencing, it is possible that novel and highly heterogeneous variants of ASF could be found, where it may be necessary to adjust the epsilon value to classify ASFV into a greater or lesser number of Biotypes. Further, as many ASFV strains, such as the isolates that make up genotype 23, ETH/AA (KT795353), ETH/017 (KT795355), ETH/1 (KT795354), ETH/004 (KT795356), ETH/2a (KT795358), and ETH/1a (KT795359), have only been partially sequenced and could not be analyzed, the number of Biotypes may expand as more historic isolates are fully sequenced.”

  1. Results, 3.1: Please specify the number of the full-length ASFV genome sequences

Answer: We have included the total number of full-length ASFV genomes to this section.

  1. Figure 3: The letters are nonreadable. Suggest increasing font size.

Answer: We have submitted a larger version of the image.

  1. Some References are not well-organized. For example, a ref is lacking in the Introduction, para 1, line 12; ref style is inappropriate in Introduction, para 5, line 7 and para 6, line 7; note in the second line below Table 2; a blank line is in ref 13 in References. 

Answer: We have corrected the formatting errors.

Reviewer 2 Report

Comments and Suggestions for Authors

In this manuscript titled “Reclassification of ASFV into 7 Biotypes Using Unsupervised Machine Learning”, the authors presented a new classification of ASFV based on comparisons performed considering the entire encoded proteome. The results reported that all ASFV genomes known to date can be classified into 7 distinct Biotypes. However, there exist many minor problems in this manuscript, which need further revision and improvement. The specific amendments are as follows:

1.         The format of the article needs attention.

2.         In the abstract, “p72” before “gene” needs to be italicized.

3.         In the introduction, “B646L” before “gene” needs to be italicized.

4.         There's some punctuation to keep in mind, such as the comma after “Dominican Republic and Haiti”.

5.         In the fifth paragraph of the introduction, the use of references should be noted: [18] [19] [20] [21{Forth, 2019 #565] should be written as [18-21], and [22] [23] should be written as [22, 23].

6.         “Materials and Methods”, “Results” and “Discussion” should all be preceded by a serial number.

7.         In section 2.2, “open reading frames” should be deleted, as already mentioned.

8.         In section 2.3, [24] [34] should be written as [24, 34].

9.         At the beginning of the third paragraph of section 3.2, what does “Error! Reference source not found.3” mean?

10.     At the end of the second paragraph of Section 3.2, the extra brackets after (Table S7)) should be deleted.

11.     In paragraph 3 of “ASFV can be Classified as 7 Biotypes”, what does “a ~6.5 kbp” mean? This needs to be checked for errors.

12.     The number before "ASFV can be Classified as 7 Biotypes" is incorrectly numbered, it should be changed to "3.3".

13.     The number before “Webportal for Automatic ASFV Biotyping and Genotyping” is incorrectly numbered, it should be changed to "3.4".

14.     In paragraph 3 of the discussion, what does “Error! Reference source not found..” mean?

15.     In paragraph 4 of the discussion, “Liv_5_40” should be written as “Liv 5/40”.

16.     In the last paragraph of the introduction, it Should be checked if “MUSCLE{Edgar, 2004 #772}” is correct.

17.     The format of reference 3 should be noted.

18.     The format of reference 14 should be noted.

19.     The format of references 33 and 35 should be noted.

20.     There are some references should be attention, such as missing page numbers for 6, 7, 9, 10, 11, 19, 21, 23, 37 and 39.

-----

some added comments:

1.         The research categorized all known ASFV genomes to date into 7 distinct biotypes.

2.         The topic is relevant to the field and addresses a specific gap in the field.

3.         The research proposes a new classification of ASFV based on a comparison of the entire coding proteome and classifies all known ASFV genomes as being classifiable into seven distinct biotypes.

4.         The data in this manuscript were selected from 220 genomes, and more data need to be selected to argue for the results.

5.         The conclusions are consistent with the evidence and arguments presented and address the main issues raised.

6.         The references in the manuscript are appropriately chosen, but attention needs to be paid to the citation format.

7.         The formatting of tables 2 and 3 needs to be consistent. The data in table 3 looks a bit confusing and needs to be adjusted.

Author Response

  1. The format of the article needs attention.
  2. In the abstract, “p72” before “gene” needs to be italicized.

Answer: The gene that encodes p72 is B646L. To maintain simplicity, consistency, and adhere to the word limit of the abstract, we exclusively refer to the terms p72 and p72 gene in the abstract.

  1. In the introduction, “B646L” before “gene” needs to be italicized.

Answer: We have corrected all instances of this formatting error.

  1. There's some punctuation to keep in mind, such as the comma after “Dominican Republic and Haiti”.

Answer: We have corrected this error

  1. In the fifth paragraph of the introduction, the use of references should be noted: [18] [19] [20] [21{Forth, 2019 #565] should be written as [18-21], and [22] [23] should be written as [22, 23].

Answer: We have corrected the references.

  1. “Materials and Methods”, “Results” and “Discussion” should all be preceded by a serial number.

Answer: We have corrected these errors

  1. In section 2.2, “open reading frames” should be deleted, as already mentioned.

Answer: We have corrected this error.

  1. In section 2.3, [24] [34] should be written as [24, 34].

Answer: We have corrected the references.

  1. At the beginning of the third paragraph of section 3.2, what does “Error! Reference source not found.3” mean?

Answer: We have corrected this error to correctly read as “Figure 3).

  1. At the end of the second paragraph of Section 3.2, the extra brackets after (Table S7)) should be deleted.

Answer: We have corrected this error.

  1. In paragraph 3 of “ASFV can be Classified as 7 Biotypes”, what does “a ~6.5 kbp” mean? This needs to be checked for errors.

Answer: We have clarified this sentence.

  1. The number before "ASFV can be Classified as 7 Biotypes" is incorrectly numbered, it should be changed to "3.3".

Answer: We have corrected this error.

  1. The number before “Webportal for Automatic ASFV Biotyping and Genotyping” is incorrectly numbered, it should be changed to "3.4".

Answer: We have corrected this error.

  1. In paragraph 3 of the dsiscussion, what does “Error! Reference source not found..” mean?

Answer: We have corrected this error.

  1. In paragraph 4 of the discussion, “Liv_5_40” should be written as “Liv 5/40”.

Answer: We have corrected this error.

  1. In the last paragraph of the introduction, it Should be checked if “MUSCLE{Edgar, 2004 #772}” is correct.

Answer: We have corrected the references.

  1. The format of reference 3 should be noted.

Answer: The format has been corrected

  1. The format of reference 14 should be noted.

Answer: The format has been corrected

  1. The format of references 33 and 35 should be noted.

Answer: The format has been corrected

  1. There are some references should be attention, such as missing page numbers for 6, 7, 9, 10, 11, 19, 21, 23, 37 and 39.

Answer: There are no page numbers for these references as they are online journals.

-----

some added comments:

  1. The research categorized all known ASFV genomes to date into 7 distinct biotypes.

Answer: Correct

  1. The topic is relevant to the field and addresses a specific gap in the field.

Answer: Correct

  1. The research proposes a new classification of ASFV based on a comparison of the entire coding proteome and classifies all known ASFV genomes as being classifiable into seven distinct biotypes.

Answer: Correct

  1. The data in this manuscript were selected from 220 genomes, and more data need to be selected to argue for the results.

Answer: The 220 genomes represent the entirety of complete non-redundant ASFV genomes.

  1. The conclusions are consistent with the evidence and arguments presented and address the main issues raised.

Answer: Thank you

  1. The references in the manuscript are appropriately chosen, but attention needs to be paid to the citation format.

Answer: We have fixed all citation problems.

  1. The formatting of tables 2 and 3 needs to be consistent. The data in table 3 looks a bit confusing and needs to be adjusted.

Answer: The headers in Table 2 have been emphasized in bold font and we have added additional borders in Table 3 at the request of the reviewer.

Reviewer 3 Report

Comments and Suggestions for Authors

The manuscript is very interesting and well prepared. A few minor questions and suggestions are included within the manuscript text.

Author Response

included within the manuscript text.

These sentences are a bit confusing and require rewording:

“There have been many reports of derivatives of the genotype 2 strain originated in Georgia, which has been isolated in Europe, Asia, Hispaniola, and in Africa [5-10]. Indeed, some of these strains have mutations across the genome, genetic rearrangements, and deletions.”

Answer: We have rewritten these sentences to read as:

Numerous reports have documented the existence of variations stemming from the genotype 2 strain whose origin can be traced back to Georgia. Genotype 2 variants have been identified in various regions across the globe, including Europe, Asia, Hispaniola, and Africa

Check agreement with title and other tables and figures - 6 or 7? :

“We discovered that there were not 25 genotypes as previously reported, and after correcting some sequence analysis errors, we established a new criteria for p72 genotyping, demonstrating the existence of only 6 genotypes.”

Answer: Here is some clarification since it can be confusing since the two numbers are nearly identical. In this manuscript we describe a method to classify full length ASFV genomes, based on their predicted encoded proteins, into 7 distinct Biotypes. In a previous study, we classified ASFV into 6 genotypes based on the amino acid sequence of a single protein, p72. Historically, ASFV genotyping was based on the partial nucleotide sequence of -72 and, without clear parameters, led to ASFV being classified into 25 different genotypes.

Reviewer 4 Report

Comments and Suggestions for Authors

This is an interesting study that applied next-generation methodology to classify the highly complex ASFV. There are not much comments, rather small questions to be reflected in the manuscript.

- 3.3 Webportal for automatic ASFV biotyping and genotyping: for Swine Fever Genomics -> "African" Swine Fever Genomics

- If authors describe why p72 match result is shown together in the webportal authors provide, it would be better to understand, because authors explained enough why not p72-sole-dependent classification works.

Author Response

#Reviewer 4

This is an interesting study that applied next-generation methodology to classify the highly complex ASFV. There are not much comments, rather small questions to be reflected in the manuscript.

- 3.3 Webportal for automatic ASFV biotyping and genotyping: for Swine Fever Genomics -> "African" Swine Fever Genomics

- If authors describe why p72 match result is shown together in the webportal authors provide, it would be better to understand, because authors explained enough why not p72-sole-dependent classification works.

Answer: Even though our proposed methodologies exhibit greater sensitivity in predicting Biotypes compared to p72 genotyping, the latter remains a widely employed technique for categorizing ASFV. Moreover, it provides a straightforward means of comparing historical strains with more recent isolates. We have clarified this in the paragraph.

Round 2

Reviewer 1 Report

Comments and Suggestions for Authors

I am satisfied with the modifications made by the authors and have no further critical comments.

Author Response

Thank you for accepting our manuscript with no further changes